# Plasma-Assisted Synthesis of TiO_2_/ZnO Heterocomposite Microparticles: Phase Composition, Surface Chemistry, and Photocatalytic Performance

**DOI:** 10.3390/molecules30163371

**Published:** 2025-08-13

**Authors:** Farid Orudzhev, Makhach Gadzhiev, Magomed Abdulkerimov, Arsen Muslimov, Valeriya Krasnova, Maksim Il’ichev, Yury Kulikov, Andrey Chistolinov, Ivan Volchkov, Alexander Tyuftyaev, Vladimir Kanevsky

**Affiliations:** 1Smart Materials Laboratory, Dagestan State University, 367000 Makhachkala, Russia; 2Joint Institute for High Temperatures, Russian Academy of Sciences, 125412 Moscow, Russia; 3Complex of Crystallography and Photonics, National Research Centre “Kurchatov Institute”, 119333 Moscow, Russia

**Keywords:** low-temperature argon plasma, titanium oxide, zinc oxide, heterocomposite microparticles, anatase, rutile, photocatalysis, solar and UV radiation, methylene blue photodegradation

## Abstract

The search for a simple, scalable, and eco-friendly method for synthesizing micro-sized photocatalysts is an urgent task. Plasma technologies are highly effective and have wide possibilities for targeted synthesis of novel materials. The mass-average temperature of plasma treatment is higher than the stability temperature of anatase and brookite, the most photoactive polymorphs of titanium dioxide. In this work, by optimizing the plasma treatment conditions and selecting source materials, a method for synthesizing micro-sized photocatalyst based on heterocomposite TiO_2_/ZnO particles with high anatase content is proposed. The synthesis method involves treating a powder mixture of titanium and zinc by low-temperature argon plasma under atmospheric conditions. The relationship between the structural-phase composition, morphology, and photocatalytic properties of the microparticles was investigated. A model for the synthesis of composite microparticles containing anatase, rutile, and heterostructural contact with zinc oxide is proposed. The photocatalytic degradation of methylene blue and metronidazole was studied to evaluate both sensitized and true photocatalytic processes. The metronidazole degradation confirmed the intrinsic photocatalytic activity of the synthesized composites. Additionally, the features of photocatalysis under UV and solar irradiation were studied, and a photocatalysis mechanism is proposed. The synthesized micro-sized heterocomposite photocatalyst based on TiO_2_/ZnO contained anatase (36%), rutile (60), and brookite (4%) and showed a photocatalytic activity during the methylene blue degradation process under UV irradiation (high-pressure mercury lamp, 250 W): 99% in 30 min.

## 1. Introduction

The rapid growth of manufacturing, pharmaceutical, and agricultural industries, which extensively use chemicals in their technological processes, has led to a severe decline of the ecological environment. The most significant impact is prominent in aquatic ecosystems [1,2].

During water filtration, the minimum particle size that can be retained is approximately several tens of microns. For molecular organic contaminants, photocatalytic purification is the most effective method [3,4,5]. The photocatalytic purification method relies on the degradation of organic contaminants under the impact of light using photoactive catalysts and is indispensable due to the inexhaustible nature of sunlight. Titanium dioxide TiO_2_ Degussa P25, composed of 75% anatase and 25% rutile phases, with particle sizes up to 20 nm is the standard commercially produced catalyst today [6,7].

The main disadvantages of TiO_2_ Degussa P25 include its high cost, toxic manufacturing involving chloride-containing compounds, and high efficiency only under UV irradiation. Moreover, such particles demonstrate rather high toxicity [8]: as their size decreases to tens of nanometers, their ability to penetrate human organs increases significantly. While some studies report no observable harmful effect of nanoparticles on the human body [9,10], more comprehensive complex research is required to confirm this. The alternative methods for TiO_2_ nanoparticle synthesis, including thermal degradation, laser ablation, and electrolysis, exist to avoid harmful manufacturing processes; however, such methods are rarely commercially viable.

To enhance catalytic activity in the visible light spectrum, heterocomposite TiO_2_-based photocatalysts have been proposed [11]. By combining TiO_2_ with other materials, such as metals [12,13] or other oxides [14,15], it is possible to extend light absorption into the visible range and reduce charge carrier recombination.

In our previous work [16], we developed an effective synthesis method for Ti/TiN/TiON/TiO_2_ composite core–shell nanoparticles exhibiting photocatalytic activity under visible and solar light irradiation. This approach is based on low-temperature electric arc plasma-assisted treatment of titanium microparticles in an open atmosphere. Plasma-chemical synthesis methods are characterized by high speed of chemical reactions and environmental friendliness. The low-temperature thermal quasi-equilibrium plasma, with the mass-average temperature reaching up to 10^4^ K, is an intermediate state between high-temperature and cold plasma types [17,18]. It can be used as a universal heat carrier and reagent and therefore offers significant technological advantages.

In our study [16], controlled phase formation was achieved through high-temperature chemical reactions of oxidation and nitridation of metallic titanium. However, the resulting photocatalytic efficiency remained relatively low despite visible-light activity, which we attribute to the presence of rutile monophase in TiO_2_. Previous studies [19] report that rutile exhibits substantially lower photocatalytic activity compared to anatase due to anatase’s better light absorption efficiency, higher mobility, and lifetime of charge carriers.

Unfortunately, monocrystalline anatase is a low-temperature phase that irreversibly transforms to rutile at 900–1000 K [20]. The minimal mass-average plasma temperature ranges from 2500 to 3000 K depending on exposure area, which far exceeds the anatase-to-rutile phase-transition temperature. Nevertheless, under “confined environment” crystallization (for example, in polymer cell) conditions, the formation of anatase mesocrystals can be possible. The first mention of mesocrystals as an intermediate state between nanocrystals and bulk materials was first introduced in [21].

According to modern classical crystallization theory, crystal nuclei form through ion attachment to a nanocluster with further formation of nanocrystal. Subsequently, these nanocrystals combine to form bulk crystals. An ordered superstructure of crystallographically oriented nanocrystalline subunits can be formed by interrupting the unification process at an intermediate stage. Such mesocrystalline state of the material significantly extends the material’s potential application range: for anatase mesocrystals, the anatase-to-rutile phase-transition temperature can increase to 1400–1500 K [22].

At approximately 1000 K, the interaction of titanium and active oxygen leads to a formation of rutile surface layer. When the temperature exceeds 2000 K, lower-valency oxides are formed between the titanium core and rutile shell. Prolonged exposure to high oxygen concentrations leads to a formation of a rutile monophase. However, under reduced oxygen supply, both phase-transition dynamics and phase formation become significantly suppressed. Thus, it is possible to create confined crystallization conditions to form anatase mesocrystals within a rutile matrix. Anatase mesocrystals then stabilize while cooling the sample. Anatase-to-rutile phase transition involves an 8% reduction in a crystalline cell volume [23], while the presence of a strong chemical bonding at an anatase–rutile intercrystalline boundary should be an additional barrier to phase transition.

To reduce the processing temperature of titanium microparticles, pure argon plasma was used without nitrogen additives [24]. Introducing zinc microparticles into the initial mixture decreases the concentration of active oxygen adsorbed on a titanium microparticle surface. Given the zinc evaporation temperature of 1180 K, the zinc subsequently evaporates, oxidizes, and covers exposed titanium particles upon entering the arc discharge gap. This approach makes it possible to effectively slow oxidation and crystallization processes in titanium and synthesize photocatalytic TiO_2_/ZnO heterocomposites. Such binary TiO_2_-ZnO [25,26,27] and other [28,29,30,31,32] systems have been extensively studied and demonstrate significant promise as photocatalysts.

This study examines the plasma-assisted synthesis of TiO_2_/ZnO heterocomposite microparticles with high anatase phase content and investigates the correlation between their phase-structural composition and photocatalytic performance. The TiO_2_/ZnO heterocomposite microparticles were synthesized through low-temperature argon plasma treatment of titanium and zinc microparticle mixture under atmospheric conditions.

## 2. Results

### 2.1. Theoretical Calculations of Velocity and Heating Temperature of Particles in Plasma

The particle heating in plasma was calculated according to [33](1)$$\Delta H\;\cdot ={\;k}_{g}\cdot T_{g}\cdot \sqrt{\frac{b}{V_{g}\cdot {µ}_{g}}\cdot }\frac{Nu}{\sqrt{\phi_{s}}}\cdot \frac{\pi }{3}\cdot D_{p}^{2}\cdot \sqrt{\rho_{p}}\;,$$
where *κ_g_* and *μ_g_*—heat conduction and viscosity of plasma; *D_p_* and *ρ_p_*—diameter and density of particle material; *Nu* and *Φ_S_*—Nusselt number and the correction to the Stoke’s drag coefficient; *T_g_* and *V_g_*—temperature and speed of plasma; *b*—length of the plasma treatment zone. The ratio $\frac{Nu}{\sqrt{\phi_{s}}}$ weakly depends on the plasma jet parameters and, under atmospheric pressure, varies in the range 2.6–2.9 for argon. Values *κ_g_*~1 Wm^−1^K^−1^ and *μ_g_*~10^−4^ kg m^−1^s^−1^ at a temperature of 3000–3500 K for argon and nitrogen can be assumed to be the same [34,35,36]. With the increase in temperature, the specific heat capacity increases. The heating temperature was calculated for rutile phase, as titanium undergoes rapid oxidation. The heat capacity of titanium at plasma temperatures was determined using the empirical formula and was ~1.1·10^3^ J·kg^−1^·K^−1^ at 3000–3500 K. Based on calculations using Equation (1), TiO_2_ particles with sizes of 50 µm, introduced in the anode region and traversing a distance of b = 17 mm through plasma with an average mass-temperature of 3000–3500 K, reached heating temperatures of approximately 2800–3000 K. Particles smaller than 30–50 µm will be heated to temperatures above the evaporation temperature of both titanium and oxygen-containing titanium phases.

### 2.2. Study of Morphology, Elemental, and Structural-Phase Compositions of Microparticles

Plasma treatment of titanium-zinc powder mixture produced ensembles of micro- and nanoparticles (Figure 1). The microparticles exhibit a hierarchical structure, consisting of a dense spherical core with linear size of several tens of microns, surrounded by a disordered arrangement of 50–100 nm nanoparticles.

According to EDX mapping analysis (Figure 2), all the formations are predominantly composed of titanium and oxygen. Zinc distribution is fragmentary, with regions of both low and high concentration. The maximum zinc content is observed in isolated nanoparticle clusters. Quantitative EDX data (Table 1) demonstrate that the titanium-to-oxygen concentration ratio in TiO_2_ microstructures is close to stoichiometric, while the zinc content remains negligible. The post-processing annealing did not alter particle morphology, and elemental composition variation in the samples changed within an instrumental error margin.

Quantitative phase analysis was performed using High Scope Plus software v.3.0.5 and the reference intensity ratio (RIR) method. The standard error of the method under these conditions is ±3 wt.%. After the plasma treatment, only oxygen-containing phases of titanium were observed according to XRD data (Figure 2) in the metallic microparticle mixture. Zinc oxide phases could not be identified in the samples. The S_0_ sample contains (Table 2) predominantly rutile (60%) and anatase (36%), and a minor amount of brookite (4%) is also present. During the post-processing treatment (sample S_300_), brookite transformed primarily to anatase, increasing the anatase-to-rutile ratio from 0.60 to 0.61.

The lattice parameters of the as-synthesized S_0_ sample are slightly lower compared to reference values (Table 3), showing a slight increase after relaxation annealing (sample S_300_). This effect is most pronounced in the anatase phase. Furthermore, we observed the high ratio of [100] (Figure 3) anatase texture, indicative of needle-like mesocrystal formation [37,38].

The XRD results enable quantitative determination of microstrain (ε) and crystallite size (coherent scattering region) values. XRD peak broadening arises due to the lattice deformations, crystallite size, and instrumental broadening (2):(2)$$b=b_{size}+b_{strain}+b_{inst},$$
where *b*—total peak broadening, rad; *b_size_*—peak broadening due to the crystallite size, rad; *b_strain_*—peak broadening due to the lattice deformation, rad; *b_inst_*—instrumental broadening, rad.

According to the Debye–Sherrer equation, *b_size_* can be determined as [39](3)$$b_{size}=\frac{0.94\cdot \lambda }{D\cdot \cos\theta },$$
where *λ*—X-ray wavelength, nm; *D*—crystallite size, nm; *θ*—XRD peak centroid position, rad; 0.94—shape coefficient [40].

The expression for *b_strain_* is given by [41](4)$$b_{strain}=\varepsilon \cdot \tan\theta ,$$

The resulting peak broadening can be approximated neglecting the instrumental broadening, as its contribution (<0.1 µrad) is negligible. To separate and estimate the contributions of crystallite size and lattice deformations to the resulting broadening, Williamson–Hall (W-H) analysis was employed [41,42]:(5)$$b=\frac{0.94\cdot \lambda }{D\cdot \cos\theta }+\varepsilon \cdot \tan\theta ,$$

According to the calculated data (Table 4), crystallite sizes decrease during the relaxation annealing. Microstresses are present for both phases, decreasing for anatase while increasing for rutile during the annealing.

The Raman spectra of the samples are presented in Figure 4. The anatase phase exhibits characteristic modes: 1A_1g_ (515 cm^−1^), 2B_1g_ (399 and 519 cm^−1^), and 3E_g_ (144, 197, and 639 cm^−1^) [43]. Rutile is characterized by the following modes: A_1g_ (612 cm^−1^), E_g_ (447 cm^−1^), and B_1g_ (143 cm^−1^) [44]. The analysis of S_0_ and S_300_ sample spectra confirms the presence of both anatase and rutile bands. Moreover, a shift of all Raman bands toward higher frequencies is observed: for S_0_ sample, anatase modes 1A_1g_ (526 cm^−1^), 2B_1g_ (402 and 529 cm^−1^), and 3E_g_ (144, 200, and 651 cm^−1^) and rutile modes A_1g_ (625 cm^−1^), E_g_ (451 cm^−1^), and B_1g_ (143 cm^−1^) are present. After the relaxation annealing, the S_300_ sample maintains both rutile and anatase bands; however, an additional shift toward higher frequencies is observed, especially for the rutile phase: A_1g_ (628 cm^−1^), E_g_ (453 cm^−1^), and B_1g_ (143 cm^−1^). Similar shifts have been previously reported for anatase and rutile samples under external pressure [40]. Notably, the intensity of anatase bands in the S_300_ sample decreases after relaxation annealing compared to the S_0_ sample.

X-ray photoelectron spectroscopy (XPS) analysis was performed to evaluate the valence states of elements, surface defects, and chemical nature of TiO_2_/ZnO heterocomposite microparticles in both as-prepared (S_0_) and annealed (S_300_) samples. The signals corresponding to characteristic Ti 2p, Zn 2p, O 1s, and C 1s bands are clearly observed in the XPS survey spectra (Figure 5a), which confirms the presence of titanium, zinc, and oxygen in all samples. The detected carbon signal mainly originates from adsorbed atmospheric contaminants and was used as the internal calibration standard (284.8 eV). Ti 2p high-resolution spectra (Figure 5c,d) demonstrate significant differences between the S_0_ and S_300_ samples. The S_0_ sample exhibits a complex Ti 2p spectrum characterized by the double doublet with Ti 2p_3/2_ at ~459.6 eV and Ti 2p_1/2_ at ~465.9 eV positions corresponding to Ti^4+^ ions and by additional peaks at ~458.3 and ~464.5 eV indicative of Ti^3+^ ions. The presence of two titanium valence states indicates the high concentration of defects induced by plasma treatment, including oxygen vacancies and imperfect Ti-O bonds. After thermal treatment at 300 °C (S_300_ sample), the Ti 2p spectrum transforms: the intensity of Ti^3+^ peaks disappears, and only peaks typical for Ti^4+^ remain.

These results indicate structural restoration of the oxide phase due to the saturation of the surface by oxygen and partial elimination of oxygen vacancies. Thus, thermal treatment contributes to the relaxation of defect structure, stabilizing the valence state of titanium. The O 1s area of the spectra (Figure 5g,h) also demonstrates changes in the defect state of the oxide lattice. For S_0_ sample, the asymmetric O 1s peak can be deconvoluted into three sub-peaks: ~529.3 eV—bonded lattice oxygen (Ti-O), ~530.3 eV—surface hydroxyl groups, and ~532.3 eV—oxygen attributed to adsorbed H_2_O molecules and/or carbon-containing contaminants. After annealing (S_300_ sample), the O 1s peak narrows and is resolved into two main components: the main peak at ~529.6 eV and the secondary peak at ~531.5 eV. The reduction in defects and hydroxyl states contribution demonstrates a reduction in the oxygen vacancies density and surface purification, which corresponds to the observed restoration of Ti valence state.

Zinc can be identified through the characteristic Zn 2p_3/2_ and Zn 2p_1/2_ (~1021.6 and ~1044 eV, respectively) doublet, with s splitting energy of approximately 23 eV, confirming the presence of Zn^2+^ in the structure (Figure 5e). This result corresponds to the formation of zinc oxide or its ultrafine derivatives near microparticle surfaces. After thermal treatment, a weakly pronounced shoulder emerges in the low-energy region of the spectrum (Figure 5f), which may be due to either surface interactions between Zn^2+^ and the titanium oxide matrix or with the partial reduction in zinc ions due to local surface bond restructuring during annealing.

Overall, the XPS analysis results confirm the high structure defectiveness characteristic of plasma-assisted synthesis and show that thermal treatment effectively reduces oxygen vacancy density. The annealing process stabilizes both surface chemistry and valence states of the main elements, which can significantly influence the material’s photocatalytic properties.

The additional information on electronic structure modifications of TiO_2_/ZnO heterocomposites electron structure after thermal treatment can be obtained by the valence band (VB) spectra analysis (Figure 5b). The VB upper edge for the S_0_ sample is located at ~1.69 eV below the Fermi level, demonstrating a significant number of defect states. These states may be associated with oxygen vacancies, Ti^3+^ ions, and crystalline disorders introduced during plasma-assisted synthesis.

These defect states generate additional electron energy levels within the band gap, obscuring the true VB position and distorting the electron distribution. After a thermal treatment (S_300_ sample), the energy of the VB upper edge is at ~0.21 eV below the Fermi level, which is the initially unexpected value for titanium oxide. However, such substantial energy shift toward the Fermi level indicates significant modifications in the surface electronic structure.

Eliminating defects (particularly Ti^3+^ states) and saturating the surface with oxygen removes intermediate energy levels within the band gap. At the same time, formation of ZnO-containing heterostructures, potential zinc surface accumulation, and possible conductivity type conversion to p-type (due to the appearance of acceptor states) can additionally shift the Fermi level toward the VB. The observed decrease in the distance between VB and the Fermi level after thermal treatment indicates the modification of the material’s electronic properties due to both surface defects elimination and interphase interactions between heterostructure components.

The photocatalytic activity of synthesized TiO_2_/ZnO heterocomposites was assessed through methylene blue (MB) aqueous solution degradation under various light sources. Figure 6 presents MB degradation curves in the presence of as-synthesized S_0_ sample, annealed S_300_ sample, and control samples, e.g., zinc-free TiO_2_ (Ref), obtained by a similar method and the catalyst-free photolysis experiment.

Prior to conducting photocatalytic tests, we investigated sample adsorption capacity during dark equilibrium stage, defined as MB concentration decrease after 30 min of stirring in the dark. The obtained results show that the thermally treated S_300_ sample demonstrates a higher degree of adsorption (~20%) compared to S0 sample (~10%).

The enhancement of the adsorption capacity after thermal treatment can be explained by surface chemical composition and structure modifications detected by XPS analysis. Specifically, the narrowing of O 1s peak and disappearance of defect-related components and oxygen vacancies indicate surface oxygen saturation and probable formation of hydroxyl and polar functional groups. Such functional groups exhibit high affinity for organic dyes and thereby enhancing MB adsorption on the S_300_ surface. Furthermore, the reduction in surface contaminations and defective states can lead to improved accessibility to active sites.

Under UV–vis irradiation (250 W high-pressure mercury lamp) conditions, the S_0_ sample demonstrated high photocatalytic activity, achieving ~99% MB discoloration within 30 min. The S_300_ sample showed slightly lower activity, reaching ~89% degradation over the same time. The control zinc-free sample Ref. showed a significantly lower activity—around ~72%—confirming zinc’s contribution to enhancing the photocatalytic properties of composites. At the same time, the catalyst-free photolysis resulted in only ~38% MB degradation, demonstrating an insignificant contribution of direct photochemical MB decomposition under the same conditions.

The reaction rate constants for the MB photocatalytic degradation were calculated using the pseudo-first-order kinetic model. As shown in Figure 6c, the highest reaction rate was observed for the S0 sample under UV–vis irradiation, reaching 0.14 min^−1^. This value is approximately five times higher than the rate constant measured for photolysis (0.027 min^−1^), indicating the high catalytic efficiency of the material. Under simulated solar irradiation, the reaction rate for the S0 sample was 0.016 min^−1^, which is more than three times greater than the rate observed for photolysis (0.005 min^−1^).

To further evaluate the photocatalytic efficiency of the synthesized material, a study was conducted to investigate the influence of the organic dye concentration and the catalyst mass on the degradation degree of MB under UV irradiation. The experimental data (Table 5) show that increasing the catalyst mass at a fixed initial MB concentration (2.5 mg/L) leads to an increase in degradation efficiency. When using 20 mg of catalyst, nearly complete degradation of the dye (99%) was achieved, whereas reducing the catalyst mass to 10 mg and 5 mg resulted in decreased degradation efficiencies of 94.5% and 86.5%, respectively. The initial concentration of the dye also had a significant effect on the resulting degradation efficiency. Increasing the MB concentration from 2.5 to 5 mg/L (at a constant catalyst mass of 20 mg) led to a decrease in degradation efficiency from 99% to 92.9%. Further increasing the MB concentration to 10 mg/L resulted in a more pronounced decline in degradation efficiency to 61.5%.

To quantitatively evaluate the efficiency of catalyst utilization, the specific photocatalytic activity was calculated as the ratio of the degradation degree of MB to the mass of the catalyst. The calculation was performed using the following expression:(6)$$Specific\;activity=\frac{Degradation\;degree,\;\;\%}{Catalyst\;mass,\;mg}$$

The results showed that as the catalyst mass decreased, the specific activity increased. This indicates that at lower catalyst loadings, each milligram of the material is used more efficiently. When the catalyst is in excess (20 mg), a portion of its surface likely remains underutilized, as the irradiation is not evenly distributed, and the overlapping of light-absorbing regions may occur. In this case, although the total degradation remains high, the material utilization efficiency is reduced.

Therefore, increasing the catalyst mass improves the overall degradation rate but does not always provide maximum efficiency per unit mass. These results suggest that selecting the appropriate catalyst dosage should involve balancing the total reaction yield with the efficient use of the catalytic material.

Additional experiments under simulated solar illumination conditions were conducted to estimate the photocatalytic activity (Figure 6b). In this case, the efficiency of all catalysts decreased, consistent with the lower photon energy contribution. However, both S_0_ and S_300_ samples maintained photocatalytic activity, achieving 65% and 58% MB degradation, respectively, whereas the catalyst-free photolysis resulted in only ~23% degradation.

To further validate the intrinsic photocatalytic activity of the synthesized TiO_2_/ZnO heterocomposites and to eliminate the influence of possible photochemical sensitization, additional photocatalytic degradation experiments were conducted using metronidazole (MNZ) as the target molecule. Unlike methylene blue, MNZ does not exhibit photochemical self-sensitization under UV or visible light. The degradation tests were performed under the same conditions as previous experiments, using both UV and simulated solar irradiation. The results are summarized in Figure 7.

Under UV irradiation, the S_0_ sample demonstrated approximately 80% degradation of MNZ within 30 min, while the S_300_ sample achieved around 75% degradation (Figure 7a).

The control catalyst-free experiment (photolysis) resulted in only ~52% degradation. Under simulated solar irradiation, photocatalytic efficiency decreased but remained significant: the S_0_ and S_300_ samples achieved MNZ degradation levels of approximately 22% and 20%, respectively, while the photolysis control showed only ~10% degradation (Figure 7b). These results confirm that photocatalytic degradation proceeds even for molecules that are not susceptible to direct photochemical decomposition under the applied light sources, thus providing solid evidence for the intrinsic photocatalytic activity of the synthesized materials.

## 3. Discussion

In our previous work [16], we reported that particle velocities through the arc discharge gap were 50–60 m/s when using argon-nitrogen mixture as a working gas, resulting in them heating only to the melting temperature. There was also no full oxidation observed, and the core–shell structure consisting of titanium core and rutile shell formed. According to calculations using (1), particles with size below 30–50 µm should overheat and evaporate. However, the presence of formed spherical TiO_2_ microparticles with sizes below 10 µm (Figure 1), indicates an overestimation of the calculated temperature.

In general, plasma’s effect on particles depends on exposure time governed by the plasma-forming gas flow rate. In this regard, the argon flow rate was decreased from 1.5 g/s to 1 g/s, reducing velocity to 30–40 m/s. Besides this, for efficient heat transfer from plasma to particles, the evaporation should be reduced to a minimum. In our case, the mixture consists of particles with different evaporation temperatures: zinc evaporates at a temperature below the mass-average plasma temperature in the processing zone, while titanium and rutile require nearly 3000 K to evaporate. Considering the shape effect, we can consider that 1–2 µm titanium microparticles would also evaporate.

During the evaporation and reaction in the oxidizing atmosphere, zinc and titanium atoms may form zinc titanates at an intermediate stage, which then decompose above 1800–2000 K into titanium and zinc oxides. The presence of zinc titanates, structurally similar to TiO_6_ octahedra [43], contributes to the transformation of low-temperature phases into rutile while reducing the transition temperature. This suggests that the spherical nanoparticles clusters consist of structural TiO_2_/ZnO units, with TiO_2_ in a rutile phase.

Microanalysis results (Figure 2) support this interpretation, as the highest zinc concentration is observed in the spherical particle region. Although zinc and titanium ionic radii of accordingly 0.60 and 0.74 pm could enable substitution of Ti^4+^ ions by Zn^2+^ ions, the absence of lattice expansion (Table 3) indicates zinc likely incorporates into the TiO_2_ lattice and probably forms an amorphous or ultrafine phase with low crystallinity. This makes it impossible to identify zinc by XRD methods in our samples.

Regarding spherical microparticles, the multi-stage nature of titanium melting and evaporation should be considered. While titanium’s melting point is 1941 K, a self-sustaining combustion of titanium initiates above 700 K, producing a massive release of nanosized TiO_2_. The presence of compact powder particles after plasma treatment suggests partial suppression of this combustion process. This occurs because active oxygen is introduced into argon plasma jet solely through diffusion from the atmosphere, maintaining low oxygen concentrations.

Besides this, the actively oxidating zinc atoms and previously evaporated titanium atoms serve as effective channels for active oxygen runoff. The oxidation reactions of titanium and zinc release an additional energy, which is localized in the reaction zone, causing localized heating.

The surface to core heating of microparticles leads to a sequential formation of low-temperature and high-temperature of TiO_2_ phases. Initially, also due to the presence of zinc impurities, the surface of spherical microparticles is covered by a rutile shell. Underlying layers, predominantly composed of anatase and brookite, partially transform to rutile with a corresponding lattice volume reduction, while the compressing central stresses inhibit pore formation. Overall, transition from anatase to rutile is a reconstruction process, accompanied by the destruction and restructuring of bonds. Such a radical restructuring of the crystal structure is rather time-dependent.

The combined XRD (Table 3 and Table 4) and Raman scattering spectroscopy (Figure 4) data confirm the formation of metastable states. Both rutile and anatase phases exhibit microstrains in the synthesized samples. These microstrains relax in anatase and increase in rutile after annealing at 300 °C.

In the as-synthesized state (sample S_0_), lattice parameters of anatase are lower compared to reference values, characteristic for polymorphous high-pressure phases. Relaxation annealing induces crystallite size reduction, which is particularly pronounced for anatase. Suggestively, strained anatase structures break down into smaller domains with dynamically changing lattice parameters. This is evidenced by the abnormal increase in anatase’s lattice parameter c after the relaxation annealing: during the anatase-to-rutile transition, parameter c undergoes the greatest transformation.

The Raman scattering spectra (Figure 4) shows a sharp decrease in anatase bands intensity after the relaxation annealing. At the same time, XRD data (Table 2) reveals an insignificant decrease in anatase content. This may be the evidence of a deeper location of anatase crystallites within the structure. Zinc incorporation, detectable by surface-sensitive methods and not by XRD, may also play a significant role.

During phase TiO_2_ phase-transition processes, impurities are excluded from the crystalline structure and accumulate at intercrystallite boundaries. Notably, the brookite phase disappears, while the anatase proportion increases slightly. Apparently, a strongly bonded composite structure of anatase–rutile is formed at intercrystalline boundaries, which contributes to anatase consolidation. The observed shift in anatase-to-rutile transition temperature relates to the presence of nanocrystals, and in analogy to [33,34], we can assume the presence of [100]-oriented anatase mesocrystal chains within the composite structure.

The results of photocatalytic tests demonstrate high efficiency of the synthesized TiO_2_/ZnO heterocomposites for MB degradation under both UV–vis and solar irradiation conditions. The plasma-treated as-synthesized S_0_ sample exhibited the highest activity, achieving ~99% MB degradation within 30 min under UV–vis irradiation conditions and significantly exceeding both zinc-free (Ref. sample) and catalyst-free (photolysis) tests. The high activity of S_0_ and S_300_ samples compared to the Ref. sample results from the presence of a significant anatase content, which demonstrates its higher photocatalytic efficiency compared to rutile.

The enhanced activity of the S_0_ sample arises from the combination of structural and electronic factors. First, the S_0_ sample structure maintains numerous surface defects, including oxygen vacancies and Ti^3+^ ions, as verified by XPS analysis. These defects introduce intermediate energy levels within the band gap and serve as traps for photogenerated electrons and holes, contributing to their spatial separation and inhibiting recombination. Thus, the defectiveness of the structure positively influences photocatalytic performance.

Second, ZnO promotes TiO_2_/ZnO heterojunction formation. Similar bandgap widths and offset alignments of valence and conduction gaps enables efficient charge carrier transfer across the phase boundary [27,28]. Such a cascade mechanism of electrons and holes transfer extends the photogenerated charge carriers’ lifetime, thereby improving the quantum efficiency of the photocatalytic process.

After thermal treatment at 300 °C, a slight photocatalytic activity reduction is observed, although a relatively high MB degradation degree (~89%) remains. This decrease may be contributed to the partial surface defect relaxation, including Ti^3+^ elimination and oxygen vacancies reduction. These changes result in an acceptor states density decrease and in a diminishing of charge transfer efficiency. However, the maintained high photocatalytic activity after the thermal treatment demonstrates the stability of the formed heterostructure and the effectivity of interphase charge carrier transfer.

The control sample Ref., containing solely the rutile phase without ZnO heterojunction, exhibited significantly lower photocatalytic activity (~72%). This performance difference highlights the crucial role of anatase/ZnO heterojunctions in enhancing light absorption and charge transfer efficiency. Furthermore, catalyst-free photolysis resulted in only ~38% MB degradation, confirming a minor contribution of direct photochemical dye degradation under given irradiation type and verifying the essential photocatalytic function of the synthesized materials.

To elucidate the reactive oxygen species (ROS) and charge carriers involved in MB degradation, radical-trapping experiments were performed using isopropanol (IPA, ^•^OH scavenger), p-benzoquinone (BQ, ^•^O_2_^−^ scavenger), EDTA (h^+^ scavenger), and AgNO_3_ (e^−^ scavenger). As shown in Figure 8, the degradation efficiency of MB after 30 min under UV irradiation decreased from 98.8 % (no scavenger) to 40.3 % with IPA, indicating a dominant role of hydroxyl radicals in the photocatalytic process. Addition of BQ and EDTA resulted in moderate suppression (80.2 % and 83.5 %, respectively), suggesting the superoxide radicals’ and photogenerated holes’ contribution, though secondary. The presence of AgNO_3_ had only a minor effect (96.7 %), implying that photogenerated electrons play a less crucial role in initiating degradation. These findings align with the literature precedents showing that ^•^OH radicals are often the main active species in heterogeneous photocatalytic degradation, while h^+^ and ^•^O_2_^−^ have supporting roles.

Experimental studies of the photocatalytic degradation of MB revealed the presence of two distinct kinetic regimes, each characterized by specific limiting factors that determine the overall process efficiency:When using a catalyst mass of 5 mg, the MB degradation efficiency decreases to 86.5%, indicating a transition to a regime limited by the availability of the active sites. In this regime, the number of adsorption centers and photocatalytically active sites on the catalyst surface becomes insufficient. Reducing the catalyst mass proportionally decreases the specific surface area available for substrate adsorption and correspondingly lowers the concentration of photogenerated reactive oxygen species. Under these conditions, despite an adequate photon flux, the reaction rate is primarily controlled by the limited number of accessible catalytic centers, which prevents complete degradation of the organic dye;When the initial MB concentration is increased to 10 mg/L (using 20 mg of catalyst), the degradation efficiency drops to 61.5%. In this case, the limiting factor shifts to the competition for photon absorption between the dye molecules and the photocatalyst. A higher dye concentration significantly increases the optical absorption of the solution, which leads to the shielding of photocatalytic particles and a reduction in excitation efficiency. Additionally, at elevated substrate concentrations, the catalyst surface becomes saturated with adsorbed dye molecules, which further enhances competition for active sites and restricts the rate of the heterogeneous catalytic process.

These findings demonstrate the complex interdependence between process parameters and provide a scientific basis for the rational optimization of photocatalytic systems to achieve maximum degradation efficiency of organic pollutants.

Under simulated solar irradiation (metal-halide lamp) conditions, all samples showed reduced but substantial MB degradation efficiency. The S_0_ and S_300_ samples maintained significant activity (~65% and ~58%, respectively), while catalyst-free photolysis resulted in only ~23% degradation. These results confirm the synthesized heterostructures’ ability to effectively use photon energy in the visible light spectrum range, which is crucial for practical solar-driven applications.

The observed high photocatalytic efficiency in MNZ degradation further supports the intrinsic photocatalytic performance of the synthesized heterostructures. The significant degradation of MNZ under both UV and solar irradiation, compared to photolysis controls, clearly excludes the possibility that the observed effects are solely due to dye sensitization, which is characteristic for MB. Unlike MB, MNZ does not act as a photosensitizer, making it a reliable probe molecule for true photocatalytic assessments. This additional evidence strengthens the validity of our conclusions regarding the photocatalytic potential of TiO_2_/ZnO heterocomposites.

For the stability assessment, we selected the TiO_2_/ZnO heterocomposite synthesized via plasma treatment without subsequent annealing (sample S_0_), as it demonstrated the highest photocatalytic activity in MB degradation. This performance is attributed to its high surface defectiveness, the presence of Ti^3+^ states and oxygen vacancies, and the efficient interfacial charge transfer within the TiO_2_/ZnO heterostructure. To evaluate reusability, five consecutive photocatalytic cycles under UV irradiation were carried out. As shown in Figure 8b, the degradation efficiency decreased only slightly from 98.8% in the first cycle to 92.6% in the fifth, confirming the high structural and catalytic stability of the material.

To assess the efficiency and prospects of the plasma-synthesized TiO_2_/ZnO heterocomposite microparticles developed in this study, a systematic comparative analysis was performed against previously reported TiO_2_–ZnO photocatalysts of various morphologies, phase compositions, and synthesis methods (see Table 6).

The literature data demonstrate that TiO_2_/ZnO photocatalysts exhibit high activity across a range of morphologies—including nanonets, nanotubes, hierarchical and inverse opal structures, and core–shell systems. In most cases, high degradation rates are achieved using relatively large catalyst dosages (0.5–2 g/L), low pollutant concentrations (1–10 mg/L), and mainly under UV or intense simulated solar irradiation. A principal distinguishing feature of our catalyst is its micron-sized (tens of microns) spherical particles with nanostructured surfaces, which sets it apart from the majority of existing nanomaterial-based powders, fibers, or films. The micron scale greatly facilitates catalyst separation and recycling, reduces material loss during operation, and enables easier scale-up for practical water treatment applications.

The TiO_2_/ZnO microparticles synthesized in this work achieve nearly complete degradation of methylene blue (up to 99%) within 30 min at a minimal catalyst dosage (1 g/L, 20 mg in 20 mL) and moderate pollutant concentrations (2.5–5 mg/L), performing as well as or better than most known analogues. Notably, their activity remains high even as the pollutant concentration increases to 10 mg/L, whereas many published systems show sharply decreased efficiency under increased contaminant loads. The multiphase composition (anatase/rutile/brookite with ZnO) and defective surfaces promote efficient charge separation and confer high material stability, as evidenced by successful degradation of both dyes and pharmaceutical contaminants (e.g., metronidazole). Moreover, the proposed low-temperature plasma-based synthesis provides a scalable and reproducible route to catalyst fabrication. It should be noted, however, that some reports describe hierarchical or inverse opal photocatalysts with comparable or even superior activity at lower catalyst dosages or under purely visible light. Other studies highlight exceptional catalyst stability due to immobilization or specific morphology. It is also important to recognize that, as with most systems, the efficiency of our material may decrease under extremely high pollutant concentrations due to active site saturation and surface shielding.

Overall, the plasma-synthesized TiO_2_/ZnO heterocomposite microparticles developed here combine high photocatalytic activity, broad applicability to various pollutants, structural and phase complexity, stability, and scalability. These features make them a highly competitive alternative to state-of-the-art photocatalytic materials for water purification, as confirmed by comparative analysis with the literature.

## 4. Materials and Methods

A mixture commercial high-purity titanium and zinc microparticles (mass ratio 10:1) was used as the initial material (previously, titanium powder was used and investigated in [16]). The metal mixture was processed near the anode region (sample S_0_) of a plasma generator featuring a plasma torch with self-aligned arc length, vortex stabilization, and an expanding output electrode channel operated under open atmospheric conditions [54]. The treatment parameters included an argon mass flow rate of 1 g/s, transporting gas mass flow rate of 0.2 g/s, powder feed rate of 10 g/min, and arc current of 150 A. A subset of resulting material was subjected to a post-treatment strain relaxation annealing in air at 300 °C for 1 h using a Naberterm tubular furnace (sample S_300_).

Microscopic studies were conducted using a JCM-6000 (JEOL, Tokyo, Japan) scanning electron microscope (SEM) equipped with an energy dispersive X-ray (EDX) microanalyzer. X-ray diffraction (XRD) patterns were obtained on an Empyrean diffractometer (PANalytical, Almelo, The Netherlands) in Bragg–Brentano geometry with CuKα radiation (λ = 1.54 Å). Surface chemical analysis of the samples was performed via X-ray photoelectron spectroscopy (XPS) using a SPECS spectrometer (Specs, Berlin, Germany). The samples were pre-etched with using 4 keV Ar^+^ ions.

The photocatalytic performance of the samples was evaluated by a degree of methylene blue (MB) photodegradation in an aqueous solution (2.5 mg/L). The photocatalytic experiments were conducted in 50 mL glass beaker under UV–visible and simulated solar irradiation conditions. A 250 W high-pressure mercury lamp (Phillips, Amsterdam, The Netherlands) without cutoff filters was used as a UV–vis light source, and a 75 W metal-halide lamp (Osram, Munich, Germany) was used to simulate solar light conditions. A constant temperature was maintained at 26 °C through ventilation and thermometer monitoring. To initiate the photocatalytic reaction, 20 mg of photocatalyst were added to 20 mL of MB aqueous solution.

Prior to illumination, the beaker was maintained in darkness for 30 min to establish an adsorption–desorption equilibrium. The suspension was subjected to ultrasonic treatment to degas the photocatalyst before conducting the experiment. Throughout the process, continuous mixing was provided by a magnetic stirrer. The light source was positioned 10 cm above the beaker. At 15-min intervals, 5 mL probes were extracted and filtered through Nylon syringe filters with pore sizes of 220 nm to separate the microparticles. The concentration of MB was quantified using an SF-2000 spectrophotometer (OKB Spektr ltd, Saint Petersburg, Russia) based on the characteristic MB absorption peak at 663.7 nm wavelength. After measuring, the solution was poured back into the reactor, and the process continued. For comparison, the MB solution was tested under similar conditions without using a photocatalyst (photolysis). The MB concentration was determined using the Beer–Bouguer–Lambert law. Additionally, to verify the intrinsic photocatalytic activity of the synthesized materials and to exclude the contribution of dye photosensitization, photocatalytic degradation tests were performed using metronidazole (MNZ) 2.5 mg/L as a model compound, which is not prone to photochemical self-sensitization. The experiments were conducted under the same conditions and with the same light sources as those used for MB degradation. The residual MNZ concentration was determined spectrophotometrically at 319 nm. Control catalyst-free photolysis tests were also performed for comparison.

## 5. Conclusions

In this paper, a simple method for the mass synthesis of a heterocomposite photocatalyst based on titanium and zinc oxides is proposed. The plasma synthesis of heterocomposite TiO_2_/ZnO microparticles with a high content of anatase phase was investigated. The synthesis involves processing a mixture of titanium and zinc in a low-temperature argon plasma under atmospheric conditions. The relationship between the structural-phase composition, morphology, and photocatalytic properties of the microparticles was investigated. A model for the synthesis of composite microparticles containing anatase, rutile, and heterostructural contact with zinc oxide is thus proposed. Additionally, the features of photocatalysis under UV and solar irradiation were studied, and a photocatalysis mechanism is proposed. The proposed method can be widely used in systems of deep purification of water resources from organic pollutants. Additional tests using metronidazole as a target pollutant confirmed the true photocatalytic nature of the synthesized heterostructures independently from dye sensitization effects.

This study was carried out within the state assignment of the Joint Institute for High Temperatures of the Russian Academy of Sciences (JIHT RAS No. 075-00269-25-00) and NRC “Kurchatov institute”.

## Figures and Tables

**Figure 1 molecules-30-03371-f001:**
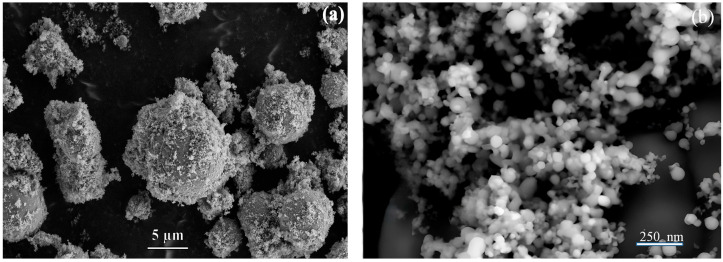
Typical SEM images of TiO_2_/ZnO (S_0_) microstructures (**a**) and nanoparticles (**b**).

**Figure 2 molecules-30-03371-f002:**
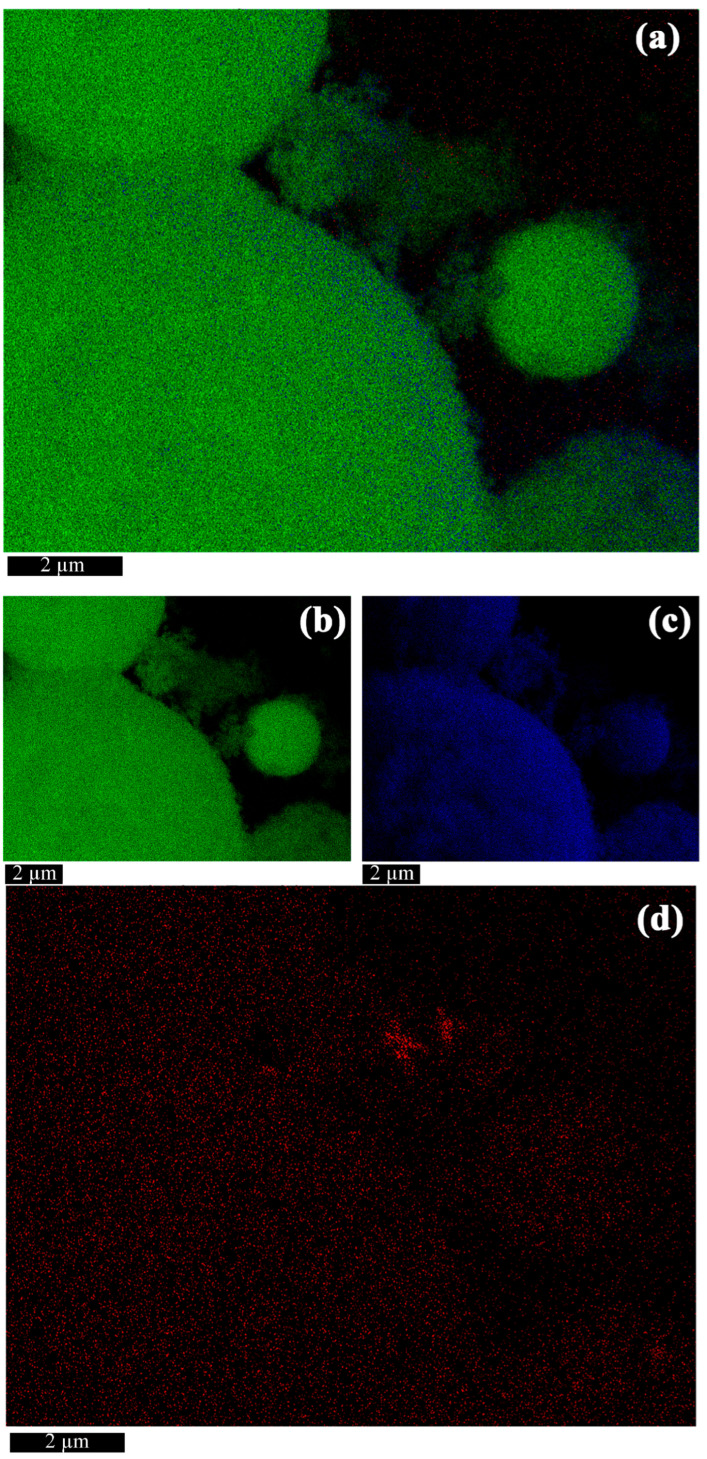
EDX mapping of TiO_2_/ZnO microstructures (S_0_). (**a**) Layered image; (**b**) Ti map; (**c**) O map; (**d**) Zn map.

**Figure 3 molecules-30-03371-f003:**
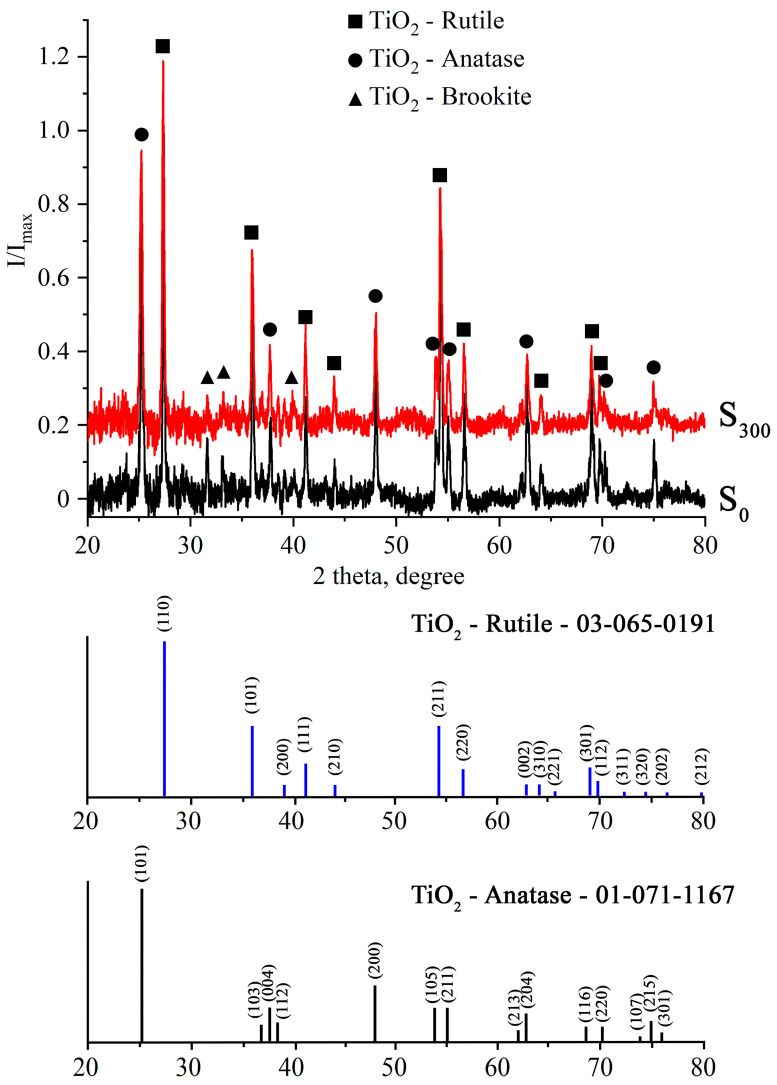
XRD patterns of TiO_2_/ZnO microstructures (S_0_ and S_300_). Designations: ■—rutile TiO_2_; ●—anatase TiO_2_; ▲—brookite TiO_2_.

**Figure 4 molecules-30-03371-f004:**
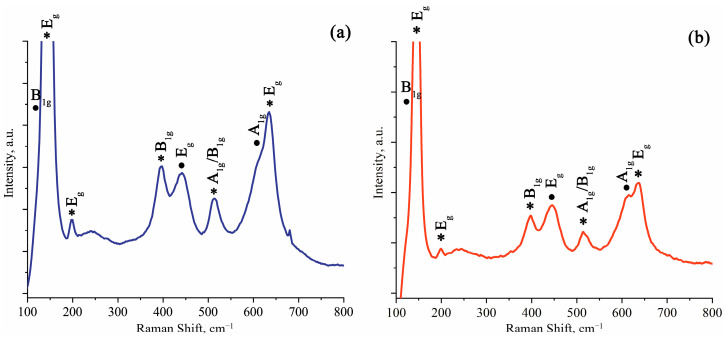
Raman scattering spectra of S_0_ (**a**) and S_300_ (**b**) samples. Designations: ●—rutile TiO_2_; *—anatase TiO_2_.

**Figure 5 molecules-30-03371-f005:**
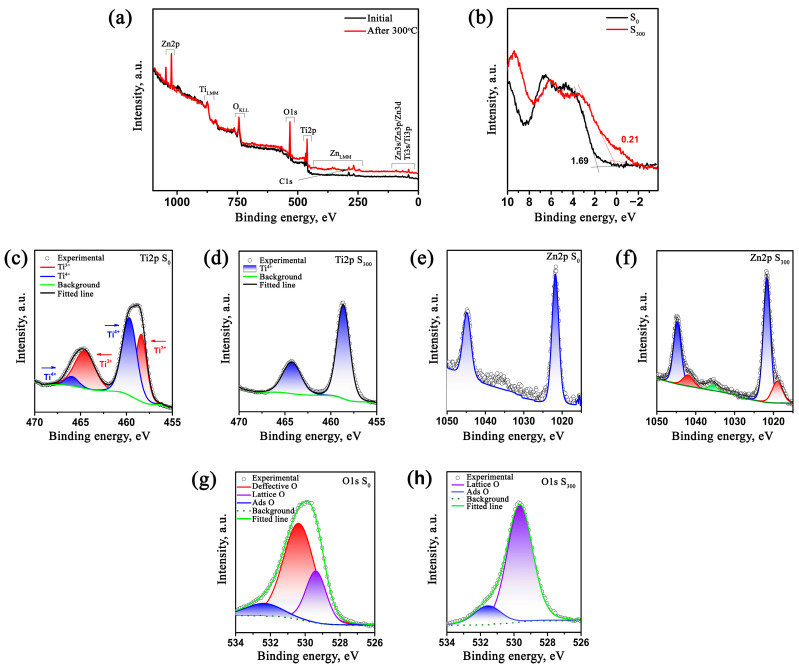
X-ray photoelectron spectroscopy (XPS) analysis of TiO_2_/ZnO microparticles before (S_0_) and after thermal treatment (S_300_): (**a**) survey spectra showing the presence of Ti, Zn, O, and C; (**b**) VB spectra; (**c**,**d**) high-resolution Ti 2p spectra indicating mixed Ti^4+^/Ti^3+^ states in S_0_ and pure Ti^4+^ in S_300_; (**e**,**f**) Zn 2p spectra; (**g**,**h**) deconvoluted O 1s spectra.

**Figure 6 molecules-30-03371-f006:**
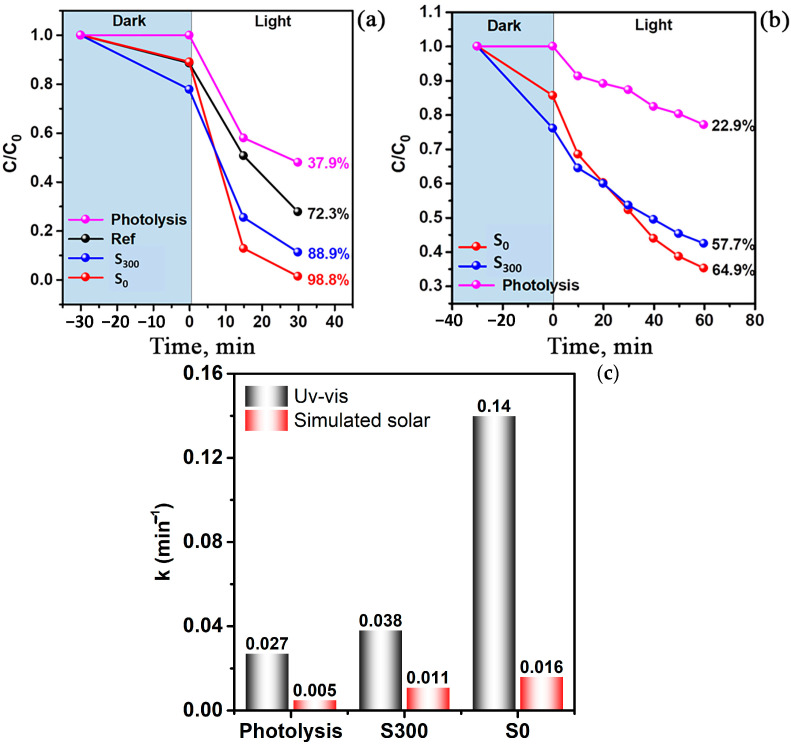
Photocatalytic degradation of methylene blue (MB, 2.5 mg/L) under different light sources using TiO_2_/ZnO microstructures: (**a**) UV–visible light irradiation (mercury lamp, 250 W); (**b**) simulated solar light irradiation (metal halide lamp, 75 W). (**c**) Reaction rate constants (k) for the photocatalytic degradation of MB calculated using the pseudo-first-order kinetic model under UV–vis and simulated solar irradiation.

**Figure 7 molecules-30-03371-f007:**
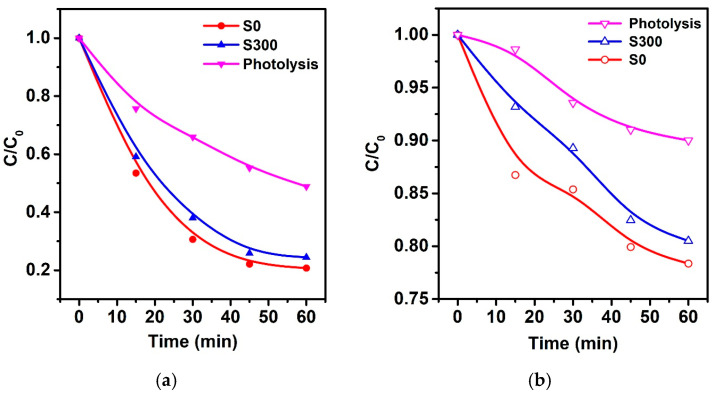
Photocatalytic degradation of metronidazole (MNZ, 2.5 mg/L) under different light sources using TiO_2_/ZnO microstructures: (**a**) UV irradiation (mercury lamp, 250 W); (**b**) simulated solar light irradiation (metal-halide lamp, 75 W).

**Figure 8 molecules-30-03371-f008:**
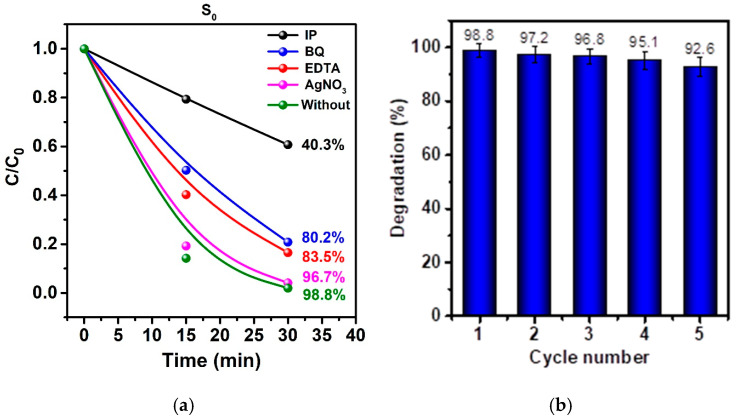
(**a**) Effect of reactive species scavengers on MB photocatalytic degradation by sample S_0_ under UV irradiation (t = 30 min). (**b**) Photocatalytic degradation efficiency of MB over five consecutive cycles using S_0_ catalyst under UV irradiation.

**Table 1 molecules-30-03371-t001:** Composition of TiO_2_/ZnO microstructures.

Type of Sample	Element	At. %
S_0_	Ti	31.41
	O	68.38
	Zn	0.21
S_300_	Ti	30.56
	O	69.26
	Zn	0.18

**Table 2 molecules-30-03371-t002:** Phase ratio in samples.

Type of Sample	Anatase, %	Rutile, %	Brookite, %
S_0_	36	60	4
S_300_	38	62	0

**Table 3 molecules-30-03371-t003:** Lattice Parameters.

Phase	Anatase		Rutile	
Lattice Parameter, Å	a	c	a	c
S_0_	3.787	9.520	4.596	2.958
S_300_	3.787	9.559	4.599	2.963
Reference	3.7892	9.5370	4.5933	2.9580

**Table 4 molecules-30-03371-t004:** Crystallite sizes and microstress values.

Phase	Anatase		Rutile	
Parameter	D, nm	ε	D, nm	ε
S_0_	72.03	0.001017	72.13	0.001351
S_300_	45.25	0.000388	64.79	0.001812

**Table 5 molecules-30-03371-t005:** Influence of catalyst mass and MB concentration on the degradation efficiency.

Experiment No.	Catalyst Mass (mg)	MB Concentration (mg/L)	Degradation Efficiency (%)	Specific Activity (% per mg of Catalyst)
1	20	2.5	99	4.9
2	10	2.5	94.5	9.5
3	5	2.5	86.5	17.3
4	20	5	92.9	4.7
5	20	10	61.5	3.1

**Table 6 molecules-30-03371-t006:** Comparative summary of composition, morphology, and photocatalytic properties of TiO_2_/ZnO-based catalysts in the degradation of organic pollutants.

Ref.	Catalyst (Composition, Ratio)	Phases/Morphology/Size	Synthesis Method	Pollutant (Type, conc.)	Catalyst Loading (Dose, Morphology, Volume)	Light Source	Reaction Time	Solution Volume	Degradation (%)
[44]	ZnO/TiO_2_ heterojunction nanomesh	Anatase + rutile TiO_2_, ZnO (zinc blend), 3D mesh, tubes 200–400 nm, particles 20–80 nm	Anodization + Zn-acetate impregnation, calc. 600 °C	Methylene blue, 5 mg/L	Mesh (200/inch), immobilized	Xe lamp 500 W (vis/UV)	90 min	50 mL	92% (ZnO/TiO_2_), 84% (TiO_2_)
[45]	ZnO/TiO_2_ nanofibers (various Zn:Ti ratios)	Hex. ZnO, anatase TiO_2_; fibers 134–228 nm; particles 31–52 nm	Electrospinning, calc. 600 °C	Methyl orange, 3 mg/L	Fibers on FTO, mass not stated	Xe, UV-A 220 W	120 min	50 mL	96% (ZnO/TiO_2_), 87% (ZnO)
[46]	IO-TiO_2_, IO-ZnO, TiO_2_/ZnO, ZnO/TiO_2_ (inverse opals, composites)	Macroporous, pores ~290 nm, film 30–40 nm	PEALD on template, annealing 500 °C	4-nitrophenol, Rh6G; conc. not specified	Coating on reactor wall	UV Osram 18 W + visible light	240 min	–	75–100% (details in text)
[47]	TiO_2_–ZnO nanocomposite (1:0.34), ~50–100 nm, polycrystal (anatase, rutile, ZnO, ZnTiO_3_, Zn_2_TiO_4_)	Anatase, rutile, ZnO, ZnTiO_3_, Zn_2_TiO_4_; 50–100 nm (TEM)	Modified sol–gel	6 antibiotics (mixture), 0.6–60 mg/L each	10 mg/L (0.01 g/L), 500 mL	Visible 125 W, UV	240 min	500 mL	>99% at 0.6 mg/L; 38–70% at 60 mg/L
[48]	TiO_2_-ZnO composite (hydrothermal), TiO_2_ nanotubes + ZnO nanoparticles	Anatase TiO_2_, hex. ZnO; nanotubes 4–5.5 µm, ZnO <100 nm	Hydrothermal	Rhodamine B, 10^−5^ M (4.8 mg/L)	Loading not specified (typically 0.5–1 g/L), 25 mL	Visible	180 min	25 mL	89% (TiO_2_-ZnO), 77% (TiO2), 31% (ZnO)
[49]	ZnO/TiO_2_ nanocomposite, sol–gel	Anatase TiO_2_, ZnO, 20–60 nm (XRD)	Sol–gel	Methylene blue, 50 mg/L	0.8 g/L, 50 mL	Visible, Xe 1000 W	120 min	50 mL	96%
[50]	TiO_2_/ZnO hierarchical fibers	Anatase TiO_2_, ZnO rods on fibers, ~100 nm	Electrospinning, hydrothermal	Rhodamine B, ~10 mg/L	Membrane, mass not specified	Visible	70 min	–	90%
[51]	Mesoporous TiO_2_–ZnO (Ti:Zn = 3:1), 10–50 nm	Anatase, wurtzite, pores 10–20 nm	Microwave hydrothermal	Doxycycline, 50 mg/L	1 g/L, 100 mL	UV 28 W (254 nm)	30–100 min	100 mL	100% (50 ppm, 30 min); 100 ppm—100 min
[52]	TiO_2_/ZnO nanocomposites (0.25–1 M ZnO), TZO1–TZO4	Anatase TiO_2_, ZnO (varied), rods 100–500 nm	Precipitation	Methylene blue, malachite green, 1 mg/L	2 g/L, 100 mL	Solar (950 W/m^2^)	60 min	100 mL	MB: up to 98% (best at 0.25 M ZnO)
[53]	ZnO–TiO_2_ composites: Z@T, T@Z (core–shell)	Anatase TiO_2_, ZnO, 30–60 nm	Precipitation + calc. 600 °C	Methyl orange, ~10 mg/L	Not specified (std. 0.5–1 g/L), 100 mL	UV	40 min	100 mL	T@Z-600: 95%, Z@T-600: 54%
This work	TiO_2_/ZnO microparticles (10:1, micron-sized), anatase/rutile/ZnO	Anatase 36–38%, rutile 60–62%, nanoparticles 50–100 nm, microspheres ~10 μm	Plasma-assisted, annealed 300 °C	Methylene blue 2.5–10 mg/L, metronidazole 2.5 mg/L	1 g/L (20 mg/20 mL), 20 mL	Hg 250 W (UV/Vis), solar 75 W	30 min	20 mL	MB: 99% (2.5 mg/L, 20 mg catalyst, 30 min); MNZ: 80%

## Data Availability

The original contributions presented in this study are included in the article. Further inquiries can be directed to the corresponding author(s).

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
