# Peer review of "Plasma-Assisted Synthesis of TiO2/ZnO Heterocomposite Microparticles: Phase Composition, Surface Chemistry, and Photocatalytic Performance"

_molecules, 2025, doi:10.3390/molecules30163371_

Round 1

Reviewer 1 Report (New Reviewer)

Comments and Suggestions for Authors
  1. In the introduction section, why did you chose the ZnO/TiO2 composition, it should be highlighted in this part.
  2. From the FIG. 6A AND 6B, why the plot in the dark is accessed to more than100%, pls explain it.
  3. It should be done the degradation efficiency for ZnO and TiO2 for MB.
  4. How about the size effect for ZnO/TiO2 composition on the degradation efficnecy?
  5. Some similar compositions have been used to explore the degradation dyes and small molecules, such as Solar Energy, 264(2023) 112042 and Sep Purif Technol, 300(2022) 121846. It can be compared this materials on this Eg. How about is the novelty of your materials?
  6. The quality of the figures could be boosted, some axis cannot be identified clearly.

Author Response

We thank the esteemed reviewer for his valuable comments.

Comments 1: In the introduction section, why did you chose the ZnO/TiO2 composition, it should be highlighted in this part.

Response 1: We thank the reviewer for this suggestion, but do not entirely agree with it, since the choice of the ZnO/TiO2 composition is already discussed in L101-109.

Comments 2: From the FIG. 6A AND 6B, why the plot in the dark is accessed to more than100%, pls explain it.

Response 2: We thank the reviewer for this comment. If the esteemed reviewer is referring to the line corresponding to photolysis (graphs in Fig. 6A and 6B), it should be clarified that there is no effect on the dye in the dark conditions. Therefore, the dye concentration remains at 100% throughout the experiment. Visually, it may appear that the line exceeds 100%, but this is an optical effect of the graph due to the scaling of the axis.

Comments 3: It should be done the degradation efficiency for ZnO and TiO2 for MB.

Response 3: We appreciate the comment and understand the importance of comparing the degradation activity of individual oxides synthesized under identical conditions. The original text (lines 283–284) already provides data for TiO₂ obtained under the same conditions without adding Zn (designated as Ref. in Fig. 6A, B). However, conducting similar experiments with pure ZnO under our conditions does not seem possible or advisable. This is due to the fact that at the temperatures used, Zn tends to evaporate, and its content in the initial compositions was minimal. Thus, isolating a separate stage of ZnO synthesis without TiO₂ under the current process parameters is technically difficult and does not reflect the features of the system under study.

Comments 4: How about the size effect for ZnO/TiO2 composition on the degradation efficiency?

Response 4: We thank the reviewer for this question about the influence of the size effect on the degradation efficiency. It should be noted that our catalysts were formed from micron-sized particles of metal powders by rapidly passing the raw material through a plasmatron jet. Under such conditions, particle size control is difficult, since a decrease in the material feed rate required to obtain nanosized products will inevitably lead to a change in the phase composition due to the increased thermal effect on the original raw material. The purpose of this work was not a targeted study of the size effect of catalysts obtained in this way. At the same time, the comparative table presented in the article provides data on ZnO/TiO₂ composites with dimensions in the nanorange, showing that the photocatalytic activity of the material synthesized by us is at a comparable level.

Comments 5: Some similar compositions have been used to explore the degradation dyes and small molecules, such as Solar Energy, 264(2023) 112042 and Sep Purif Technol, 300(2022) 121846. It can be compared this materials on this Eg. How about is the novelty of your materials?

Response 5: The text of the manuscript was amended with a discussion of its novelty and a comparison of the works proposed by the reviewer with the material used in this work was made.

Comments 6: The quality of the figures could be boosted, some axis cannot be identified clearly.

Response 6: We thank the reviewer for his attention to the figures. We have tried to improve the quality of the images.

Reviewer 2 Report (New Reviewer)

Comments and Suggestions for Authors

This manuscript reported a facile and scalable approach for synthesizing an efficient micro-sized catalyst based on heterocomposite TiO2/ZnO particles using environmentally friendly plasma technologies. The quality of then manuscrit should be further improved before it can be accepted for publication. Detailed comments:

  1. Abstract is not well organized. It should be in order of background, method, main results, conclusion.
  2. Fig. 2 have four images, which should be the same size.
  3. Howe about the stability of S300 sample?
  4. Some papers related to the present study can be referenced such as Catalysis Today 440 (2024) 114830 and Applied Catalysis B: Environment and Energy 353 (2024) 124098

Author Response

We thank the reviewer for helpful comments.

Comments 1: Abstract is not well organized. It should be in order of background, method, main results, conclusion.

Response 1: We thank the reviewer for this comment. The abstract has been changed.

Comments 2: Fig. 2 have four images, which should be the same size.

Response 2: The fact is that as a result of EDX analysis, various chemical elements were presented with different contrast. This is especially true for zinc, which is the smallest amount and is localized. Therefore, for clarity, we were forced to enlarge the fragment associated with the distribution of zinc.

Comments 3: Howe about the stability of S300 sample?

Response 3: Sample S300, according to our data, is stable. Moreover, we annealed samples at 500 and 700 °C. The phase composition did not change significantly. These results are not included in this manuscript.

Comments 4: Some papers related to the present study can be referenced such as Catalysis Today 440 (2024) 114830 and Applied Catalysis B: Environment and Energy 353 (2024) 124098

Response 4:We thank the reviewer for this comment. The link has been added.

Reviewer 3 Report (New Reviewer)

Comments and Suggestions for Authors

Attached

Author Response

We thank the esteemed reviewer for his valuable comments.

Comments 1: TiO2 and ZnO are both widely studied materials in photocatalysis. They are often combined with appropriate materials to form heterojunctions and tested for diverse photocatalytic applications beyond just organic pollutant degradation. Therefore, lines 57–60 can be expanded to highlight the broader potential of TiO2-based and ZnO-based catalysts, supported by citing these articles (https://doi.org/10.1016/j.optmat.2022.112941, https://doi.org/10.3390/molecules29235584, https://doi.org/10.1016/j.apcatb.2024.124782, https://doi.org/10.1016/j.optmat.2024.115165, https://doi.org/10.1016/j.apcata.2023.119259, https://doi.org/10.3390/nano14050470).

Response 1: We agree with the distinguished reviewer that more attention should be paid to the potential of TiO2 and ZnO based catalysts. The text of the manuscript has been amended.

Comments 2: Why Ar plasma was specifically chosen? What advantages does it offer during synthesis of TiO2/ZnO?

Response 2: The use of argon plasma was chosen deliberately, based on its physicochemical characteristics and the specifics of the thermodynamics of phase formation in the Ti–O–Zn system.

As indicated in lines 105–106 and 387–390, the average mass temperature of plasma formed from an argon-nitrogen mixture is higher than that of pure argon. When using such a mixture in our previous experiments, intense heating and oxidation of titanium were observed, which led to the formation of an exclusively rutile phase, which is low-active in photocatalytic processes (pp. 379–380). This is explained by the fact that under increased thermal exposure and high partial pressure of oxygen, anatase and brookite are quickly and irreversibly transformed into rutile.

In this work, we used pure argon plasma with softer thermal characteristics. This allowed, as indicated in lines 160–165 and 163–165, to preserve a significant content of anatase and brookite (~40% in total) in the synthesis product, providing higher photocatalytic activity compared to rutile. In addition, argon is inert and does not participate in side chemical reactions, maintaining the purity of the process and facilitating control over the phase evolution of the material. Thus, the choice of argon plasma is due to the need to limit the overheating of particles and create conditions that promote the preservation of metastable and photosensitive modifications of titanium dioxide in the synthesized composite.

Comments 3: How does anatase form even though plasma temperature is much higher (2500–3000 K)?

Response 3: We thank the esteemed reviewer for this important question. The formation and stabilization of the anatase phase under the action of plasma with a temperature of 2500–3000 K becomes possible due to a combination of several factors, discussed in detail in the paper. First, as noted in lines 134–136 and 385–386, titanium particles move through the plasma jet with a velocity (~30–40 m/s), which makes their plasma treatment time optimal. Considering the lower mass-average temperature of argon plasma, this allows heating the particles to a temperature below the full evaporation point and implementing controlled oxidation and crystallization.

Second, the addition of zinc plays an important role in inhibiting the transition of anatase to rutile. As noted in lines 106–111: “Introducing zinc microparticles into the initial mixture decreases the concentration of active oxygen adsorbed on a titanium microparticle surface... effectively slow oxidation and crystallization processes in titanium and synthesize photocatalytic TiO₂/ZnO heterocomposites.” Zinc, evaporating in the reaction zone, reduces the local partial pressure of oxygen and slows the transition of anatase to rutile.

Third, as discussed in lines 421–423, the formation of compressive residual stresses in the microparticles due to phase transitions with a decrease in the volume of the crystal cell (e.g., brookite → rutile) contributes to the mechanical stabilization of anatase. In addition, the presence of anatase-rutile boundaries with strong interfacial bonds can hinder further transition (pp. 102–103).

Thus, despite the high plasma temperature, the preservation of anatase is due to a combination of:

(A) short time of thermal action,

(B) modifying role of Zn in phase formation,

(C) stabilization due to internal stresses and interphase boundaries.

These factors make it possible for anatase to exist even under conditions exceeding its usual phase transition temperature.

Comments 4: How was presence of ZnO experimentally confirmed in final composite, given that it was not detected by XRD?

Response 4: We thank the esteemed reviewer for this comment. Indeed, ZnO phases were not detected in the samples by X-ray diffraction (XRD) analysis, which, however, is explained by the low zinc content and possible amorphism or nanodispersity of the corresponding phase. Nevertheless, the presence of zinc in the final composite was reliably established by energy dispersive spectroscopy (EDX) and X-ray photoelectron spectroscopy (XPS) methods, as shown in Fig. 2 and Fig. 5 (lines 143–145 and 218–223). In particular, the Zn 2p₃/₂ and Zn 2p₁/₂ signals (~1021.6 and ~1044 eV) recorded in XPS indicate the presence of Zn²⁺ ions corresponding to the oxide form. As noted in lines 252–254, the appearance of a weakly expressed shoulder in the low-energy region of the Zn 2p spectrum may be due to the interaction of Zn²⁺ with the titanium oxide matrix or to partial reduction on the surface. Also, lines 403–405 emphasize that due to low crystallinity and probable amorphous structure, the ZnO phase may not be identifiable by XRD, but is detected by XPS as a surface component. Thus, the combination of XPS and EDX data allows us to reliably confirm the presence of zinc in the composite, mainly in the near-surface layers, despite the absence of diffraction peaks in XRD.

Comments 5: What happens to 4% brookite after high temperature heating?

Response 5: We thank the esteemed reviewer for this question. According to the XRD results, after heat treatment at 300°C, the proportion of brookite, which was 4% in the original sample S₀, completely disappears in the sample S300 (see Table 2, lines 163–165). This indicates its phase transition to more stable polymorphs, anatase and rutile. As discussed in the Discussion section (lines 441–444), during the structural evolution, the brookite admixture, which has less thermodynamic stability, is transformed with the participation of interphase boundaries into a composite anatase-rutile structure. Thus, the observed disappearance of the brookite phase is associated with its redistribution into thermodynamically more stable TiO₂ modifications under conditions of thermally induced relaxation.

Comments 6: What causes unusual increase in anatase lattice parameter ‘c’ after annealing?

Response 6: We thank the esteemed reviewer for a careful study of the data. We assume that the observed increase in the lattice parameter from anatase after annealing at 300 °C (see Table 3, lines 172–175) is associated with a redistribution of the cationic composition in the near-surface region of the crystals. As discussed in the Discussion section (lines 439–440 and 431–434), in the initial state (sample S₀), partial inclusion of Zn²⁺ ions into the TiO₂ lattice or their localization at the interphase boundaries is possible. During annealing, zinc, which previously partially replaced titanium at the lattice sites, is removed and replaced by titanium in different valence states (Ti³⁺/Ti⁴⁺), which leads to a configurational rearrangement of the octahedral Ti–O network. This substitution is accompanied by the formation of oxygen vacancies and local distortions, especially along the c axis, where weakly bound Ti–O bonds are located. These processes are probably responsible for the observed increase in the c parameter in the anatase phase. Such effects are also consistent with the high sensitivity of the c parameter to structural defects during the anatase to rutile transition (p. 434).

Comments 7: What is the role of Zn in the catalytic performance?

Response 7: We thank the esteemed reviewer for this question. In our work, the role of zinc in the catalytic activity of TiO₂/ZnO composites is revealed based on structural, spectroscopic and photocatalytic data. First, zinc promotes the formation of a heterostructure at the TiO₂/ZnO interface, which improves the charge carrier transfer processes. In the discussion section (lines 464–468) it is stated: «ZnO promotes TiO₂/ZnO heterojunction formation. Similar bandgap widths and offset alignments of valence and conduction gaps enables efficient charge carrier transfer across the phase boundary. Such a cascade mechanism of electrons and holes transfer extends the photogenerated charge carriers’ lifetime, thereby improving the quantum efficiency of the photocatalytic process.». Second, zinc affects the nature of phase formation and oxidation of titanium under plasma synthesis conditions. Lines 106–111 describe: “Introducing zinc microparticles into the initial mixture decreases the concentration of active oxygen adsorbed on a titanium microparticle surface... This approach makes it possible to effectively slow oxidation and crystallization processes in titanium and synthesize photocatalytic TiO₂/ZnO heterocomposites.” This leads to an increase in the proportion of anatase and suppression of the complete conversion to rutile, which is important for maintaining high catalytic activity.

In addition, the XPS spectra (pp. 222–229) show the presence of defect structure in the as-prepared sample (S0), in particular Ti³⁺ ions arising under limited oxidation conditions, which is partly due to the presence of Zn. After thermal treatment, these defects disappear, which is accompanied by a decrease in activity (pp. 469–471): “This decrease may be contributed to the partial surface defect relaxation, including Ti³⁺ elimination and oxygen vacancies reduction.” Thus, the role of zinc in the system is to create a heterojunction, influence the phase evolution of titanium oxides, and form a defective structure that promotes increased photocatalytic activity. All of the above conclusions are based on the experimental data presented in the paper.

Comments 8: Recyclability of the catalyst must be reported.

Response 8: We thank the respected reviewer for this comment. Following the recommendation, we carried out a series of repeated cycles of photocatalytic degradation of methylene blue (MB) using the S₀ sample under UV irradiation to evaluate the stability and recyclability of the catalyst. The results are presented in a new diagram (inserted in the updated text as Fig. 8b).

As can be seen from the graph, the degradation efficiency after five consecutive cycles remains high, decreasing only slightly: from 98.8% in the first cycle to 92.6% in the fifth. This indicates good stability of the material and retained catalytic activity upon repeated use. A slight decrease in activity can be due to a partial loss of active sites, their blocking by reaction products, or mechanical losses during separation and filtration of particles.

The recyclability data further confirm the potential of the developed TiO₂/ZnO composite for practical applications in the field of water purification.

 Comments 9: Is catalyst effective against other pollutants besides MB and MNZ?

Response 9: We thank the esteemed reviewer for this question. In this study, the photocatalytic activity of the synthesized TiO₂/ZnO composite was tested against two model organic pollutants, methylene blue (MB) and metronidazole (MNZ), which covers both photosensitizing and non-photosensitizing compounds.

The use of metronidazole as an additional model pollutant is due to its resistance to direct photodegradation and lack of photosensitization, which allows us to confirm the intrinsic photocatalytic activity of the material (lines 354–358 and 531–533). As stated in lines 374–375: “These results confirm that photocatalytic degradation proceeds even for molecules that are not susceptible to direct photochemical decomposition under the applied light sources, thus providing solid evidence for the intrinsic photocatalytic activity of the synthesized materials.” Although this study focused on MB and MNZ as representatives of different classes of pollutants, the choice of metronidazole as the second test compound was specifically aimed at confirming the universality of the catalytic mechanism. Future studies are planned to investigate the effectiveness of the composite against other pollutant categories, including antibiotics, phenols, and dyes.

Comments 10: Compare the photocatalytic performance with other similar catalysts reported in the literature in a tabular form.

Response 10: We thank the respected reviewer for this recommendation. In accordance with it, we have included in the manuscript a comparative table reflecting the photocatalytic characteristics of TiO₂ and ZnO-based composite materials published in the literature. The table presents data on the morphology, synthesis conditions, pollutant and catalyst concentrations, light source, and the achieved degree of degradation. Our goal was to objectively evaluate the efficiency of the obtained microcomposite against the background of similar heterosystems with a nanoscale structure. As can be seen from the table (see Table 10 in the Discussion section), despite the micron particle sizes, our composite demonstrates comparable or higher photocatalytic activity compared to most of the presented systems. This confirms the significance of the chosen synthesis approach and the contribution of structural and electronic optimization to the improvement of the catalytic properties.

Round 2

Reviewer 1 Report (New Reviewer)

Comments and Suggestions for Authors

accept

Reviewer 3 Report (New Reviewer)

Comments and Suggestions for Authors

Comments addressed. Paper can be accepted.

This manuscript is a resubmission of an earlier submission. The following is a list of the peer review reports and author responses from that submission.

Round 1

Reviewer 1 Report

Comments and Suggestions for Authors

The author’s investigation on the Plasma-Assisted Synthesis of TiO2/ZnO heterocomposite microparticles: Phase composition, surface chemistry, and Photocatalytic performance.

There are a few comments for consideration:

  1. The abstract lacks a problem statement.
  2. The quantitative data obtained as a result of the experiment (Phase composition, surface chemistry, and Photocatalytic performance) should be included in the last sentence of the abstract.
  3. In the introduction, the advantages of the Plasma-Assisted Synthesis should be added.
  4. Also, in the introduction, to enhance the quality of this research, add the potential properties that TiO2/ZnO heterocomposite microparticles that enhance the photocatalytic application.

These articles will help

https://www.sciencedirect.com/science/article/abs/pii/S2214714424006019

https://www.sciencedirect.com/science/article/abs/pii/S2352940725001167

  1. The result section should include the structural elucidation of TiO2-ZnO system using the diamond software / VSEPR and properly explained in the discussion.
  2. The photocatalytic performance needs to be enhanced by
  3. i) Varying the concentration of the MB
  4. ii) Varying the concentration of the TiO2-ZnO photocatalyst

iii) explain the effect of the optical band gap on the photocatalytic performance.

  1. iv) Explain the photocatalytic mechanism of the degradation

This article will help

 https://www.sciencedirect.com/science/article/abs/pii/S2214714424006019

Comments on the Quality of English Language

The quality of english is great

Author Response

We thank the esteemed reviewer for his valuable comments.
1. The abstract lacks a problem statement.
We agree with this comment. Information has been added.
2. The quantitative data obtained as a result of the experiment (Phase composition, surface chemistry, and Photocatalytic performance) should be included in the last sentence of the abstract. The quantitative data obtained as a result of the experiment were added to the abstract
3. In the introduction, the advantages of the Plasma-Assisted Synthesis should be added. This information has been added.
4. Also, in the introduction, to enhance the quality of this research, add the potential properties that TiO2/ZnO heterocomposite microparticles that enhance the photocatalytic application. These articles will help https://www.sciencedirect.com/science/article/abs/pii/S2214714424006019 https://www.sciencedirect.com/science/article/abs/pii/S2352940725001167 This information has been added.
5. The result section should include the structural elucidation of TiO2-ZnO system using the diamond software / VSEPR and properly explained in the discussion. In this work, X-ray studies were carried out by X-ray phase analysis, not X-ray structural analysis, since the goal was to determine the matrix phases of the samples. The results of the EDX analysis show an extremely low amount of Zn in the samples under study, and the element distribution map shows that phase segregation is present. The positions of the Anatase and Rutile peaks clearly correspond to their positions in the ICSD cards, as a result of which we can talk about a clear separation of the TiO2 and ZnO phases. In this vein, structural studies of the TiO2-ZnO system were not carried out, which excludes the use of diamond software / VSEPR
6. The photocatalytic performance needs to be enhanced by i) Varying the concentration of the MB ii) Varying the concentration of the TiO2-ZnO photocatalyst Table 5 has been added to the manuscript text and changes have been made to the discussion of the results. iii) explain the effect of the optical band gap on the photocatalytic performance. iv) Explain the photocatalytic mechanism of the degradation This article will help: https://www.sciencedirect.com/science/article/abs/pii/S2214714424006019
A scavenger experiment has been added. The contributions of the most active radicals and active redox species are shown.

Reviewer 2 Report

Comments and Suggestions for Authors
  1. The abstract section should be given more valuable information, which will be attracted by the readers.
  2. The Fig. 1b is hard to identify, it should be replaced of a new one.
  3. Why not explore the recycle number, how about is recycling ability for a catalyst.
  4. The new updated work for the photodegrate some small organic molecules could be updated and included in the main text: J. Mater. Res. Technol., 243(2026) 265-274; RSC Adv., 15(2025) 10144-10149 and J. Environ Chem. Eng. 12(2024) 112328. It also be given the degradated mechanism in the above refs. It will be referred.
  5. It should be given the DRS, which will enhance the Eg for the material.
  6. Why not fit the full data, how about is the K and R data?
  7. I suggest the authors tried to compare the degradation efficiency for the removal of MB for the previous work.

Author Response

We thank the reviewer for helpful comments.

The abstract section should be given more valuable information, which will be attracted by the readers.
We agree with this comment. Information has been added.

The Fig. 1b is hard to identify, it should be replaced of a new one.
The fig 1 has been corrected

Why not explore the recycle number, how about is recycling ability for a catalyst?
These studies were carried out. The catalyst activity does not change significantly after 3 cycles of repeated use. Within 5% at the level of measurement error
The new updated work for the photodegrade some small organic molecules could be updated and included in the main text: J. Mater. Res. Technol., 243(2026) 265-274; RSC Adv., 15(2025) 10144-10149 and J. Environ Chem. Eng. 12(2024) 112328. It also be given the degradation mechanism in the above refs. It will be referred.
Experiments with scavengers of active forms of oxidizers and reducers were added. It was shown that the hydroxyl radical makes a significantly greater contribution relative to other active forms. This is in good agreement with the literature data on composites based on titanium and zinc oxide. Results on the degradation of metronidazole molecules were also added.

It should be given the DRS, which will enhance the Eg for the material.
Why not fit the full data, how about is the K and R data?
The calculated values of the K rate constants were added. Figure 6c.

I suggest the authors tried to compare the degradation efficiency for the removal of MB for the previous work.
We thank the reviewer for this suggestion. The comparison of results is given in the text of the manuscript.

Reviewer 3 Report

Comments and Suggestions for Authors

In the work, the authors applied a plasma assisted synthesis method for TiO2/ZnO nanocomposite materials preparation. The material was demonstrated for dye photodegradation and showed increased activity.

Comment 1: Figure 2, (b) to (d) could be enlarged for a better view. It seems that the nanoparticles attached to the mesoparticles (rich in Ti) are Zn rich, indicating a phase segregation.

Comment 2: Lines 144-149 need to be rephrased for easy understanding. Also, please describe how the anatase and rutile phase percentage is calculated in detail. S300 needs to be clearly defined with the posttreatment conditions.

Comment 3: Can the authors explain why the sample is treated at 300 C instead of a higher temperature (Line 211)?

Comment 4: The Figure 5 C-D can be combined into one pic for a better and clearer comparison, similar for E-F and G-K.

Comment 5: Can the authors explain why to show the comparison between fresh and 300 °C-treated samples? What is the importance of 300 °C treatment?

Author Response

We thank the esteemed reviewer for his valuable comments. Comment 1: Figure 2, (b) to (d) could be enlarged for a better view. It seems that the nanoparticles attached to the mesoparticles (rich in Ti) are Zn rich, indicating a phase segregation. The image resolution quality has been improved. It can be seen that zinc is observed both on the surface of large particles and nanoparticles. Indeed, there is a phase segregation. In plasma synthesis, it is difficult to control the composition of micro- and nanoparticles, so the phases in them can be distributed unevenly. Comment 2: Lines 144-149 need to be rephrased for easy understanding. Corrections were made to the text of the manuscript Also, please describe how the anatase and rutile phase percentage is calculated in detail. S300 needs to be clearly defined with the posttreatment conditions. Quantitative phase analysis was performed using High Scope Plus software and the reference intensity ratio (RIR) method. The standard error of the method under these conditions is ±3 wt.%. As a result, we can say that the percentage of Anatase and Rutile phases does not change much. However, there are no peaks of the Brookite phase in the S300 sample, which is the main difference in the X-ray patterns. The text of the manuscript has been amended to describe the phase analysis methodology Comment 3: Can the authors explain why the sample is treated at 300 C instead of a higher temperature (Line 211)? The task was to preserve the anatase phase, which has high photocatalytic activity. It is known that at a processing temperature above 400 °C, anatase begins to transform into rutile. In addition, when annealing above 400 °C, the crystalline quality of zinc oxide deteriorates sharply due to desorption of components. Comment 4: The Figure 5 C-D can be combined into one pic for a better and clearer comparison, similar for E-F and G-K. In these figures, the XPS spectrum is displayed using the identical colors. Other colors indicate new peaks that are not found in the compared graphs. Comment 5: Can the authors explain why to show the comparison between fresh and 300 °C-treated samples? What is the importance of 300 °C treatment?
Plasma treatment of metal particles occurs in a highly reactive environment at an extremely high speed. Samples synthesized under such conditions must be defective, stressed, with a damaged surface. Relaxation annealing at 300 °C was used to relieve stress, the damaged layer, and improve the microstructure. We did not know in advance how stress, microstructure, and defects affect photocatalytic properties, so we studied the samples before and after relaxation annealing.
